# Correlations between Cognitive Evaluation and Metabolic Syndrome

**DOI:** 10.3390/metabo13040570

**Published:** 2023-04-17

**Authors:** Andrei Cătălin Oprescu, Cristina Grosu, Walther Bild

**Affiliations:** 1Department of Physiology, Faculty of Medicine, “Grigore T. Popa” University of Medicine and Pharmacy, 700115 Iași, Romania; 2Department of Neurology, Faculty of Medicine, “Grigore T. Popa” University of Medicine and Pharmacy, 700115 Iași, Romania; cristina.grosu@umfiasi.ro

**Keywords:** type 2 diabetes, Alzheimer’s disease, obesity, insulin resistance, leptin

## Abstract

One of the most common medical diseases is metabolic syndrome (MetS), which encompasses diabetes and obesity. It has a systemic effect, which has long-lasting consequences on the body that are still not fully understood. The objectives of the study were to investigate the association between the severity of metabolic imbalances, insulin resistance, leptin concentration, and the presence of cognitive disorders and to assess the possible protective role of some classes of drugs used in the treatment of type 2 diabetes mellitus (T2D) and dyslipidemia in order to identify a viable target in the near future. The study included 148 diabetic patients. Standardized tests for the evaluation of cognition, including Mini-Mental State Examination (MMSE) and Montreal Cognitive Assessment (MoCA), were applied to all study participants. Serum concentrations of leptin and insulin were determined using the enzyme-linked immunosorbent assay method (ELISA), and insulin resistance was calculated using the homeostatic model assessment for insulin resistance (HOMA-IR). We found that MMSE and MoCA scores were associated with anthropometric parameters, and MoCA was associated with glycemic control parameters and leptin levels. Further research is needed in order to establish the magnitude of the relationship between metabolic syndrome components and cognitive decline in diabetic patients.

## 1. Introduction

Alzheimer’s disease (AD) is a progressive brain disorder. The National Institute on Aging defines it as a degenerative brain disorder that slowly destroys memory and thinking skills, eventually leading to the inability to carry out simple tasks [1]. The definition from the Alzheimer’s Association describes AD as a progressive and degenerative disorder that affects brain cells, leading to memory loss and changes in behavior and personality [2]. According to the *Diagnostic and Statistical Manual of Mental Disorders, Fifth Edition* (DSM-5), AD is classified as a neurocognitive disorder that involves the gradual onset and continued decline of cognitive functioning, including memory, orientation, language, judgement, and reasoning [3]. Having type 2 diabetes mellitus (T2D), a chronic metabolic illness, is known to raise the chance of AD by at least two-fold [4,5]. In 2021, 537 million people (10.5% of the population) were living with diabetes, with an estimated increase to 643 million in 2030 (11.3% of the population). It is believed that 24% of adults over the age of 75 have T2D [6]. Dementia now affects about 55 million people worldwide and will reach 78 million by 2030. Age is the main risk factor for dementia, and the world’s population is aging, so the number of people affected by a form of dementia will increase, which means declining quality of life, dependency, institutionalization, mortality, and huge social and economic costs [7]. Chronic hyperglycemia can cause neuronal damage through the formation of advanced glycation end products (AGEs), which trigger the synthesis of reactive oxygen species (ROS) and the production of pro-inflammatory cytokines, contributing to microvascular changes and systemic inflammation [5]. The Rotterdam study was the first to show an increased risk of AD in patients with T2D [8], but numerous other studies have since reported lower cognitive performance in people with T2D, compared to healthy controls of the same age [9], or a faster rate of cognitive decline than that normally associated with natural aging [10]. 

The study of global trends in diabetes since 1980 (751 studies on various populations, with 4.4 million participants) found that diabetes is a risk factor for dementia [11]. Insulin resistance characteristic of T2D has been proposed as a pathogenic factor in the onset of AD by impairing cerebral glucose metabolism, leading to neurodegeneration and cognitive impairment. The explanation can be found in peripheral hyperinsulinemia and insulin deficiency in the brain, which decreases the permeability of the blood–brain barrier to insulin, as well as insulin receptor dysfunction, low levels of glucose-3 transporters (GLUT3) and of components of the insulin-signaling pathway, and IGF-1 in the central nervous system [12]. Both T2D and AD show evidence of inflammation, oxidative stress, mitochondrial dysfunction, advanced glycation end products, and amyloid deposition [13].

Diabetes and dementia are a global challenge for screening and management. Diets that can prevent or slow down the progression of the disease are being considered [14]. Drugs that are effective in the treatment of T2D, including those which decrease insulin resistance and restore insulin-signaling pathways, by diminishing competitive inhibition on the insulin-degrading enzyme (IDE), can be used in AD (intranasal insulin, glucagon-like peptide-1 receptor agonists, peroxisome proliferator-activated receptor-gamma agonists, and others still under study) [15].

Proper glycemic control in the stages of mild cognitive impairment (MCI) can induce the reversal of these disorders or slow the progression of the disease. Unfortunately, the extent of cognitive impairment is unknown because routine screening is not performed.

In order to be able to offer patients with metabolic syndrome the best care and to be as efficient as possible, it is useful to find a pattern that is easy to apply on a large scale. It needs to be inexpensive for the health systems but able to identify high-risk people and those who already have cognitive disorders in the early stages. In the first step, these patients can benefit from treatment for metabolic pathology that will also have a positive effect on neurodegenerative pathology (ex. GLP-1 RA, SGLT2-i) [16,17]. In the second step, the cognitive decline can be slowed down if they reach the psychiatrist/neurologist in the early phases, given the fact that promising evidence has been published on a new drug that results in moderately less decline in measures of cognition and function [18].

The objectives of the study were to investigate the association between the severity of metabolic imbalances, insulin resistance, leptin levels, and the presence of cognitive disorders and to assess the possible protective roles of some classes of drugs used in the treatment of T2D and dyslipidemia in order to identify a viable target in the near future.

## 2. Experimental Design

### 2.1. Study Population

In this study, we evaluated diabetic patients from Iași County, Romania, registered at two different hospitals, who were aged over 50 and from both rural and urban environments. When included in the study, the patients were diagnosed with T2D for at least a year and were being treated with oral antidiabetic drugs (OADs), non-insulin injectable therapy, or insulin. The participants were enrolled in the study from June 2019–September 2022. The study protocol was approved by the ethics committee of the “Grigore T. Popa” University of Medicine and Pharmacy. All patients included in the study gave written informed consent and understood and signed for the acquisition, analysis, and publication of anonymized data collected during the study. In brief, patients were eligible to participate if they were aged >50 years and did not have a confirmed diagnosis of cognitive disorders, a history of stroke, or psychiatric pathology. Exclusion criteria included inability to provide informed consent, failure to provide accurate anamnestic medical data, and a history of cognitive disorder, stroke, or psychiatric pathology. 

### 2.2. Clinical Interview and Physical Examination

Patients self-reported socio-demographic parameters. Associated pathological conditions were ascertained from anamnesis and local hospital records. Waist circumference (WC), weight, and height were measured, and body mass index (BMI) was calculated with the following formula: weight divided by height square. Obesity was defined as a BMI ≥ 30 kg/m^2^, and overweight was defined as a BMI between 25 and 29.9 kg/m^2^ [19]. Chronic complications of diabetes were determined annually during routine periodic examination, and data were registered in their charts. Microvascular complications refer to diabetic retinopathy (regular eye fundoscopy performed by ophthalmologists), diabetic nephropathy (periodic determination of estimated glomerular filtration rate, albumin-to-creatinine ratio, and screening for urinary infection to rule out the false positive results), and diabetic peripheral neuropathy (foot examination and assessment of the protective sensation with 10 g monofilament, vibration perception using a 128 Hz tuning fork, a tip-therm, and a pin-prick test). Macrovascular complications refer to prior diagnoses, registered in patients’ charts, of stroke, cardiac ischemic disease, and peripheral arterial disease. 

### 2.3. Biomarker Measurement

Blood samples were collected during routine visits patients with T2D made periodically. The lipid profile included total cholesterol, low-density lipoprotein (LDL) cholesterol, high-density lipoprotein (HDL) cholesterol, and triglycerides; the glycemic profile included fasting blood glucose, glycated hemoglobin (HbA1c), and uric acid, which were measured. All samples were analyzed on the same day in the hospital laboratory. In addition, samples were collected separately for the determination of insulinemia and leptin. Shortly after collecting, the serum was separated by centrifugation. Afterwards, serum samples were stored at −20 degrees Celsius for a maximum of 2–3 months. The enzyme-linked immunosorbent assay (ELISA) method was used to measure serum leptin and insulin concentrations using Human Leptin ELISA, clinical range, Cat. No RD191001100, and kit Insulin Elisa, ref DKO075. Using serum insulin concentration and fasting blood glucose, we determined the homeostatic model assessment for insulin resistance (HOMA-IR). It was suggested that the cutoff value was ≥2 for insulin resistance [20]. 

### 2.4. Definition of Metabolic Syndrome

MetS is defined as the presence of at least 3 of 5 components: increased WC ≥80 cm in women or ≥94 cm in men; elevated triglycerides ≥150 mg/dL or treatment for elevated triglycerides; reduced HDL cholesterol ≤50 mg/dL in women or ≤40 mg/dL in men; elevated blood pressure when systolic blood pressure (SBP) ≥130 mmHg, diastolic blood pressure (DBP) ≥ 85 mmHg, or treatment for high blood pressure; and elevated fasting plasma glucose ≥ 100 mg/dL or treatment for diabetes [21]. 

### 2.5. Cognitive Examination 

Two questionnaires were used to evaluate the cognitive function, which were included in the study: Mini-Mental State Examination (MMSE) and Montreal Cognitive Assessment (MoCA). Participants underwent neurocognitive testing in a quiet hospital room. A total of 30 points were possible in both scales; a score of 26 or higher was considered normal for the MoCA score, and a value above 24 meant no cognitive impairment in the MMSE score.

The administration of the MMSE test lasted between 5 and 10 min. MMSE, or the Folstein test [22], is a screening test for impaired cognitive function in adults, fast and easy to perform, that assesses the temporo-spatial orientation, attention, immediate and short-term memory, the ability to perform concrete and abstract operations, motor skills, and language. However, because it includes language and math test items, people with poor education might have difficulties understanding. The MMSE test is sensitive to tracking cognitive decline in patients with AD, with an annual decline of 1.8 to 3.2 points in total [22].

MoCA (created in 1996 by Ziad Nasreddine in Quebec) [23] is a first-line tool that requires only 10 min to apply and tests short-term memory, executable performance, attention, concentration, and more. It was created to be a quick screening tool for mild cognitive dysfunction. It is organized in eight sections: visuospatial/executive skills (5 points), naming (3 points), memory (no points), attention (6 points), language (3 points), abstraction (2 points), delayed recall (5 points), and orientation (6 points). For people with less than 12 years of schooling, an additional point was added. 

### 2.6. Statistical Analysis

Statistical analysis was performed with the Statistical Package for Social Sciences, version 20. The Kolmogorov test was used to evaluate the normal distribution of the analyzed data. Pearson correlation coefficients were determined to assess the association between variables. To identify the clinical and metabolic determinants of cognitive scores, linear regression models were applied. 

## 3. Results

We analyzed a group of 148 patients. Of these, 50 were men (33.78%). The mean age of these patients was 67.07 ± 5.79 years (Table 1). All patients were known to have type 2 diabetes when they were enrolled in the study. The mean duration of diabetes was 8.3 ± 6.23 years (Table 1). Most patients in the study group had comorbidities, dyslipidemia, and hypertension.

The mean value of WC was 108.52 ± 10.90 cm, and for the BMI, the value was 32.31 ± 5.36 kg/m^2^ (Table 1). The majority of the patients were overweight (n = 47, 31.8%) and obese (n = 95, 64.2%). 

We analyzed the metabolic profiles of the patients included in the study, which consisted of a glycemic control outside the treatment targets (mean fasting blood glucose of 149.24 ± 41.48 mg/dL, mean HbA1c of 7.38 ± 1.2%). Additionally, regarding the lipid profiles of these patients, we found LDL-cholesterol outside the target values (100.12 ± 36.04 mg/dL) (Table 1). 

We analyzed the values of insulinemia and serum leptin in the study group. Based on the value of fasting insulinemia and blood glucose, we calculated the HOMA-IR index. This was moderately increased in the study group, thus identifying patients with insulin resistance (Table 1). 

Analyzing the results obtained in the MMSE questionnaire, we found a mean value of 25.55 ± 3.48, with a minimum score of 12 and a maximum score of 30. For the MoCA score, the mean value obtained in the study population was 20.63 ± 5.04 (Table 1). Cognitive impairment was detected in 26.4% (n = 39) of patients when using the MMSE score and in 17.7% (n = 26) when using MoCA.

Almost all patients in the study group were treated with metformin (93.9%), and about one-third of patients were treated with insulin (27%) (Table 2). The frequency of statin use was 68.9%.

More than a third of the patients in the study had microvascular complications, and almost one quarter of the patients had macrovascular complications (Table 3). 

We found a statistically significant positive correlation between the MMSE score and the level of education but not with the duration of diabetes. A statistically significant negative correlation was observed between the MMSE score and diabetes control parameters (HbA1c and glycemia). However, the MMSE score was not correlated with other parameters included in the definition of the metabolic syndrome (Table 4). 

On the other hand, the MoCA score was related to the anthropometric characteristics of the study population, with both waist circumference and BMI having a statistically significant negative correlation. 

The results also showed a negative correlation between the leptin value and the MoCA score (Table 4). 

We also found no statistically significant correlations between MMSE or MOCA scores with the lipid profile, nor with insulinemia or the HOMA-IR index (Table 4).

We analyzed the MMSE and MoCA scores based on the treatment of patients who might be affected by cognition, namely by insulin treatment and metformin treatment. We did not find statistically significant differences between the scores of patients receiving insulin treatment or metformin treatment vs. those who did not have this treatment (Table 5).

The presence of diabetes-specific complications did not influence the MMSE score or the MoCA score, as shown in Table 6.

Both scores were lower in patients with macrovascular complications; however, a statistically significant relationship was observed only in the case of the MoCA score (Table 7).

The MMSE total score was related to HbA1c (Table 4). When adjusted for age, HbA1c predicted the MMSE total score (β = −0.185, *p* = 0.02). Thus, in our sample, it was observed that a higher HbA1c value predicted a lower MMSE total score (Table 8). However, when adjusting for formal education, this relationship was not significant anymore (Table 8).

The MoCA score was negatively related to clinical and biological parameters, which included WC, BMI, SBP, HbA1c, and leptin (Table 4). This relationship was preserved for WC (β = −0.205, *p* = 0.001), BMI (β = −0.182, *p* = 0.003), and leptin (β = −0.136, *p* = 0.032) when adjusted for age and formal education (Table 9). HbA1c predicted the decrease in the MoCA score when adjusted for age (β = −0.181, *p* = 0.021) but not for formal education (β = −0.106, *p* = 0.083) (Table 9). 

## 4. Discussions 

The field of research to assess mild cognitive impairment (MCI) is still a challenging domain. There are rare longitudinal studies published that note cognitive impairments in relation to metabolic syndrome elements (insulin resistance) [24,25,26]. No test is actually diagnostic for MCI (It only assesses a degree of cognitive dysfunction.).

People with diabetes have many causes of cognitive decline (vascular and/or metabolic), and their screening should be performed periodically, as for any chronic complication. Cases with early impairment in orientation, attention, memory, language, and visual-spatial skills should be sent be to a neurologist for diagnosis (cerebrospinal fluid biomarkers, neuroimaging). For those without cognitive changes, the tests used can be a benchmark for further evolution. The pathology of neurodegenerative diseases is a continuum from the preclinical stages to the prodromal stages without obvious functional impact to dementia. Quantifying the cognitive impairment stage of MCI is a challenge because it appears to be a window of opportunity for diagnosis and treatment. There is no specific test to confirm the diagnosis of MCI [27].

The diagnosis of MCI is misused, and it should meet the criteria listed in Table 10.

In our sample, cognitive impairment was detected in 26.4% (n = 39) of patients when using the MMSE score and in 17.7% (n = 26) when using the MoCA score. In another study, MCI was observed in 38 (54.29%) patients with T2D, and normal cognition was observed in 32 (45.71%) [29]. A study conducted in our country on T2D patients, aged between 33 and 81 years, found that a percentage of 42.03% of the patients presented MCI [30].

In our study, the degree of formal education and the MMSE score showed a statistically significant positive relationship, but no relation was observed for the duration of diabetes. Diabetes management parameters and the MMSE score showed a statistically significant negative association (HbA1c and glycemia). The MMSE score, however, did not correspond with any of the other factors used to define metabolic syndrome. WC and BMI had a statistically significant negative relationship with the MoCA score, suggesting a link between the anthropometric characteristics of the study population and cognitive disfunction. In another study, performed on a group of 207 T2D patients from Timisoara, who were evaluated with MMSE and with psychological tests and neurological examination, including imaging (computerized tomography and magnetic resonance imaging), those with MCI had a mean age of 63 (57.00–71.00) years, older than patients without MCI, who had a mean age of 52.00 (45.00–61.00) years (*p* < 0.001). Duration of diabetes and body fat were also correlated with MCI, similar to other components of macroangiopathy [30]. In our study, WC, BMI, and leptin were negatively correlated with the MoCA score but not with the MMSE score. 

Many studies in this field are on small groups of patients. For instance, in one study, 30 people over 50 years old, with and without DM, were evaluated with MMSE and Modified Mini-Mental Status Examination (3MS). Diabetes was associated with lower levels of cognitive function. The correlations between age, sex, duration of diabetes, and HbA1c among diabetics with impaired cognitive status were not significant [31].

A larger cohort study on 1519 elderly people with DM grouped into three groups according to HbA1c, to which MMSE was applied, found thatHbA1c ≥ 8% was independently associated with the severity of cognitive decline [32]. In our sample, there was a correlation between the MMSE total score and HbA1c; a higher HbA1c value predicted a lower MMSE total score. Nevertheless, this association lost significance when accounting for formal schooling.

In the Irish Longitudinal Study on Aging (TILDA), the association between DM in people over 50 years of age and cognitive decline was sought in a 6-year follow-up, using MMSE and MoCA (3687 participants were evaluated). At baseline, the prevalence of diabetes was 6.4%. Participants with DM had significantly lower MoCA and MMSE scores and a higher number of errors than those without DM. Age, male sex, stroke, and hypertension were significantly associated with a higher number of errors in the MMSE score at baseline. Over six years, DM was significantly associated with an accelerated decline in cognition [33].

A previous study on 138 T2D patients showed a relationship between cognitive disfunction assessed by MMSE and MoCA scores and the BMI, HDL cholesterol, and HbA1c, similar to our results [34].

Our results showed that clinical and biochemical variables, including leptin, WC, BMI, SBP, and HbA1c were inversely correlated with the MoCA score. When age and formal education were taken into account, this link was still present for WC, BMI, and leptin. When adjusted for age, HbA1c predicted the decline in the MoCA score, but not when the formal education was used as a predictor. These results could be explained by the difference in the sensitivity and specificity of the detection of mild cognitive disfunction by these scales, as shown in different studies. A study evaluated the MoCA test, applied to 70 patients with T2D, and the correlations with HbA1c, fasting, and postprandial blood glucose, age, and duration of diabetes. Those with MCI had higher HbA1c (8.79 ± 1.85 % vs. 7.78 ± 1.60 %), higher fasting blood glucose (177.05 ± 62.48 mg/dL vs. 149.38 ± 54.38 mg/dL), and higher postprandial blood glucose (282.03 ± 85.61 mg/dL vs. 214.50 ± 82.43 mg/dL), relations which were statistically significant [29]. One pilot study compared MMSE with MoCA for the diagnosis of MCI in 30 patients with T2D (over 50 years in whom depression and dementia were excluded). The authors calculated sensitivity (MMSE 13%, MoCA 67%) and specificity (MMSE 93%, MoCA 93%), positive (MMSE 66%, MoCA 84%) and negative (MMSE 51%, MoCA 56%) predictive values, likelihood ratios (MMSE 1.8, MoCA 9.5), the Kappa statistic (MMSE 0.07, MoCA 0.4), and the area under the curve (AUC) (MMSE 0.46, MoCA 0.7). The MoCA appeared to be a better screening tool than the MMSE for MCI in the diabetic population [35]. The metanalysis conducted by Ciesielska et al. [36] suggested that the MoCA score better meets the criteria for screening tests for the detection of MCI among patients over 60 years of age than MMSE. This meta-analysis evaluated the credibility of MoCA vs. MMSE in the detection of MCI while considering sensitivity and specificity using cut-off points (20 studies for MoCA assessment and 13 for MMSE). Diagnostic accuracy for MoCA and MMSE was calculated by ROC curves. The MoCA test better fulfilled the criteria for screening tests in patients over 60 years of age than the MMSE (AUC for MoCA: 0.846, 95% CI 0.823–0.868; for MMSE: 0.736, 95% CI 0.718–0.767) [36].

A longitudinal study showed that insulin resistance, rather than single elevation of blood glucose, predicts cognitive decline, particularly memory, in people with prediabetes [37]. One review showed that metabolic syndrome contributes to the development and progression of AD; however, the factors linking this association have not been determined. IR is at the heart of metabolic syndrome and is probably the key link between metabolic syndrome and AD [38]. Another review argued (through animal studies, as well as clinical studies) that insulin resistance is a risk factor for dementia and that treatments that combat it can be beneficial [39].

These data and our results suggest that people with diabetes should be cognitively screened periodically, especially those with IR (manifested through as hyperinsulinism or increased WC).

Our study had some limitations. First of all, our sample size was relatively small, and further research is needed to establish the magnitude of the relationship between MetS components and cognitive decline in diabetic patients. Moreover, we did not include a control group consisting of non-diabetic individuals, which could have allowed us to make more inferences and comparisons on the influence of MetS components in the MCI.

## 5. Conclusions

There is no specific test to confirm the diagnosis of mild cognitive impairment (MCI). However, numerous screening tools can also be used by non-specialists, such as Mini-Mental State Examination (MMSE) or Montreal Cognitive Assessment (MoCA); these tools can help identify people with T2DM, who are already at increased risk for the development of MCI, earlier. Our study favored the use of MoCA, but further studies are warranted.

This section is not mandatory, but may be added if there are patents resulting from the work reported in this manuscript.

## Figures and Tables

**Table 1 metabolites-13-00570-t001:** Characteristics of the study sample.

Parameters	Mean	Std. Deviation	95% CI for Mean		Min.	Max.
			Lower Bound	Upper Bound		
Age (years)	67.07	5.790	66.13	68.01	53	87
School years	11.11	4.026	10.46	11.77	0	26
DM duration (years)	8.30	6.237	7.28	9.31	0	47
Weight (kg)	86.52	14.777	84.12	88.92	58	133
WC (cm)	108.52	10.901	106.75	110.29	80	135
BMI (kg/m^2^)	32.31	5.36	31.44	33.18	22.58	54.24
SBP (mmHg)	139.91	17.340	137.10	142.73	99	189
DBP (mmHg)	79.73	10.672	78.00	81.46	50	122
HbA1c	7.3823	1.20053	7.1873	7.5773	5.00	11.40
Glycemia (mg/dL)	149.24	41.486	142.50	155.98	85	333
Total cholesterol (mg/dL)	177.78	42.501	170.87	184.68	89	323
HDL chol (mg/dL)	47.344	13.9246	45.082	49.606	7.0	113.0
LDL chol (mg/dL)	100.122	36.0403	94.248	105.997	35.0	219.0
Triglycerides (mg/dL)	152.57	67.824	141.56	163.59	59	445
Uric acid (mg/dL)	5.58	1.375	5.36	5.80	2	9
Insulinemia	15.89	12.078	13.87	17.91	1	82
HOMA-IR	5.495	4.781	4.718	6.271	0.001	24.945
Leptin	27.31	27.411	22.73	31.89	1	112
Total score MMSE	25.55	3.480	24.98	26.11	12	30
Total score MoCA	20.63	5.044	19.80	21.45	5	29

**Table 2 metabolites-13-00570-t002:** Frequency of using different treatments for diabetes in the study population.

Treatment	Frequency	Percent
Insulin	40	27
Metformin	139	93.9
Sulphonylureas	18	12.2
DPP4 inhibitors	31	20.9
GLP1 receptor agonists	16	10.8
SGLT2 inhibitors	14	9.5
Other treatments	3	2

**Table 3 metabolites-13-00570-t003:** Frequency of diabetes complications in the studied sample.

Complications	Frequency	Percent
Microangiopathic	56	37.8
Macroangiopathic	39	26.4

**Table 4 metabolites-13-00570-t004:** Correlations of MMSE and MoCA scores with the studied parameters.

Parameters		Total Score MMSE	Total Score MoCA
Age (years)	Pearson Correlation	−0.284 **	−0.320 **
	Sig. (2-tailed)	<0.001	<0.001
T2Dduration (years)	Pearson Correlation	−0.051	−0.077
	Sig. (2-tailed)	0.540	0.355
School years	Pearson Correlation	0.610 **	0.648 **
	Sig. (2-tailed)	<0.001	<0.001
Weight (kg)	Pearson Correlation	0.048	−0.120
	Sig. (2-tailed)	0.559	0.148
WC (cm)	Pearson Correlation	−0.147	−0.321 **
	Sig. (2-tailed)	0.075	<0.001
BMI (kg/m^2^)	Pearson Correlation	−0.058	−0.257 **
	Sig. (2-tailed)	0.486	0.002
SBP (mmHg)	Pearson Correlation	−0.117	−0.169 *
	Sig. (2-tailed)	0.156	0.041
DBP (mmHg)	Pearson Correlation	0.042	−0.120
	Sig. (2-tailed)	0.616	0.149
HbA1c	Pearson Correlation	−0.196 *	−0.194 *
	Sig. (2-tailed)	0.017	0.018
Glycemia (mg/dL)	Pearson Correlation	−0.174 *	−0.130
	Sig. (2-tailed)	0.034	0.117
Total cholesterol (mg/dL)	Pearson Correlation	0.134	0.119
	Sig. (2-tailed)	0.104	0.152
HDL cholesterol (mg/dL)	Pearson Correlation	0.086	0.105
	Sig. (2-tailed)	0.301	0.204
LDL cholesterol (mg/dL)	Pearson Correlation	0.106	0.065
	Sig. (2-tailed)	0.203	0.434
Triglycerides (mg/dL)	Pearson Correlation	0.065	0.107
	Sig. (2-tailed)	0.429	0.197
Uric acid (mg/dL)	Pearson Correlation	0.148	0.123
	Sig. (2-tailed)	0.073	0.137
Insulinemia	Pearson Correlation	0.115	0.031
	Sig. (2-tailed)	0.175	0.720
HOMA IR	Pearson Correlation	0.089	0.009
	Sig. (2-tailed)	0.284	0.916
Leptin	Pearson Correlation	−0.033	−0.210 *
	Sig. (2-tailed)	0.699	0.013

* *p* < 0.05, ** *p* < 0.001.

**Table 5 metabolites-13-00570-t005:** Differences in MMSE and MoCA scores according to treatment.

Treatment		Mean	Std. Dev.	95% CI for Mean		p-sig.
				Lower Bound	Upper Bound	
Total score MMSE	without insulin	25.47	3.394	24.82	26.12	0.668
	with insulin	25.75	3.740	24.55	26.95	
Total score MoCA	without insulin	20.70	5.094	19.72	21.68	0.769
	with insulin	20.43	4.966	18.84	22.01	
Total score MMSE	without metformin	24.00	5.025	20.14	27.86	0.170
	with metformin	25.65	3.358	25.08	26.21	
Total score MoCA	without metformin	18.44	6.267	13.63	23.26	0.181
	with metformin	20.77	4.948	19.94	21.60	

**Table 6 metabolites-13-00570-t006:** Differences in MMSE and MoCA scores according to microvascular complications.

Diabetes Microvascular Complications		Mean	Std. Dev.	95% CI for Mean		p-sig.
				Lower Bound	Upper Bound	
Total score MMSE	without	25.62	3.331	24.93	26.31	0.747
	with	25.43	3.741	24.43	26.43	
Total score MoCA	without	20.96	4.942	19.93	21.99	0.313
	with	20.09	5.206	18.70	21.48	

**Table 7 metabolites-13-00570-t007:** Differences in MMSE and MoCA scores according to macrovascular complications.

Diabetes Macrovascular Complications		Mean	Std. Dev.	95% CI for Mean		p-sig.
				Lower Bound	Upper Bound	
Total score MMSE	without	25.87	3.356	25.23	26.51	0.058
	with	24.64	3.703	23.44	25.84	
Total score MoCA	without	21.16	4.635	20.27	22.04	0.033
	with	19.15	5.851	17.26	21.05	

**Table 8 metabolites-13-00570-t008:** HbA1c as a predictor of the MMSE total score.

Dependent Variable: Total Score MMSE	Standardized Coefficients	Sig.	95.0% Confidence Interval for B	
	Beta		Lower Bound	Upper Bound
HbA1c ^1^	−0.196	0.017	−1.034	−0.105
HbA1c ^2^	−0.185	0.020	−0.983	−0.087
HbA1c ^3^	−0.117	0.076	−0.717	0.036
HbA1c ^4^	−0.113	0.080	−0.694	0.039

^1^ unadjusted; ^2^ adjusted for age; ^3^ adjusted for formal education (years); ^4^ adjusted for age and formal education (years).

**Table 9 metabolites-13-00570-t009:** Predictors of the MoCA total score.

Dependent Variable: Total Score MOCA		Standardized Coefficients	Sig.	95.0% Confidence Interval for B	
		Beta		Lower Bound	Upper Bound
Unadjusted	Age (years)	−0.320	0.000	−0.413	−0.143
	School years	0.648	0.000	0.654	0.966
	BMI (kg/m^2^)	−0.257	0.002	−0.391	−0.093
	WC (cm)	−0.321	0.000	−0.220	−0.077
	SBP (mmHg)	−0.169	0.041	−0.096	−0.002
	HbA1c	−0.194	0.018	−1.491	−0.140
	Leptin	−0.210	0.013	−0.069	−0.008
Adjusted for age	BMI (kg/m^2^)	−0.296	0.000	−0.418	−0.138
	WC (cm)	−0.326	0.000	−0.218	−0.083
	SBP (mmHg)	−0.141	0.073	−0.086	0.004
	HbA1c	−0.181	0.021	−1.402	−0.116
	Leptin	−0.219	0.006	−0.069	−0.012
Adjusted for formal education	BMI (kg/m^2^)	−0.147	0.022	−0.255	−0.020
	WC (cm)	−0.194	0.002	−0.147	−0.032
	SBP (mmHg)	−0.058	0.365	−0.054	0.020
	HbA1c	−0.111	0.081	−0.992	0.058
	Leptin	−0.123	0.065	−0.047	0.001
Adjusted for age and formal education	BMI (kg/m^2^)	−0.182	0.003	−0.284	−0.059
	WC (cm)	−0.205	0.001	−0.149	−0.040
	SBP (mmHg)	−0.044	0.481	−0.048	0.023
	HbA1c	−0.106	0.083	−0.948	0.059
	Leptin	−0.136	0.032	−0.048	−0.002

**Table 10 metabolites-13-00570-t010:** Criteria for the identification of MCI, according to the MCI Working Group of the European Consortium on Alzheimer’s Disease, Brescia Meeting, Italy, June 2005 [28].

Cognitive complaints coming from the patients or their families
The reporting of a decline in cognitive functioning relative to previous abilities during the past year by the patient or informant
Cognitive disorders as evidenced by clinical evaluation (impairment in memory or in another cognitive domain)
Absence of major repercussions on daily life (the patient may, however, report difficulties concerning complex day-to-day activities)
Absence of dementia

## Data Availability

The data presented in this study are available on request from the corresponding author. The data are not publicly available due to privacy.

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
