# Peer review of "Correlations between Cognitive Evaluation and Metabolic Syndrome"

_metabolites, 2023, doi:10.3390/metabo13040570_

Round 1

Reviewer 1 Report

The study has addressed the potential relationship between e metabolic imbalances, IR, leptin, and cognitive disorders in diabetic patients. The study's findings suggest that there may be an association between cognitive function and metabolic imbalances, IR, and leptin levels. Additionally, the study identifies the potential drug targets that may help protect against cognitive decline in diabetic patients. However, the study's sample size is relatively small, and further research is needed to establish the magnitude of the relationship between metabolic syndrome components and cognitive decline in diabetic patients. In summary, the study sheds light on the potential association between metabolic syndrome and cognitive decline in diabetic patients, which could lead to the development of new treatment options in the future. Overall, while the study provides valuable insights into the potential relationship between metabolic syndrome and cognitive decline in diabetic patients, the limitations should be taken into account when interpreting the findings.

1.      The study did not account for other potential factors that could influence cognitive function, such as age, education level, or medication use, which could have affected the results.

2.      While the study used standardized tests for the evaluation of cognitive function, the MMSE and MoCA tests may not be sensitive enough to detect subtle cognitive changes that occur in diabetic patients.

Why did this study not include a control group of non-diabetic individuals, which limits the ability to compare the results to people without diabetes?

Reviewer 2 Report

Dear authors,

The manuscript has a lot of work to be done.

I recommend reading the suggested changes (attached below) and implementing them. 

Also, it would be beneficial for you to enlist the assistance of a certified Medical English translator in proofreading and grammatically correcting the manuscript, as the level of the English language is not up to par with the standards for scientific articles.

Lines 14 and 15 "assess the possible protective role of some classes of drugs used in the treatment of T2D and dyslipidemia," - you cannot use an abbreviation like T2D if it is not explained beforehand. The proper form would be "assess the possible protective role of some classes of drugs used in the treatment of Type 2 diabetes mellitus (T2D) and dyslipidemia

Lines 15 and 16 "The study included a number of 148 diabetic patients" - change to "The study included 148 diabetic patients". 

Lines 16 and 17 "(MMSE and MoCA)" - please write the full names of the tests before the abbreviations

Lines 17 and 18 "The circulating level of leptin and insulin was determined" - use a plural form, also use "serum concentrations" instead of "circulating levels"

Lines 27-29 "Alzheimer’s disease (AD) is an incurable, degenerative illness that slowly robs people of their ability to think, recall, and reason, making it impossible for them to rule their everyday lives" - This formulation is far too colloquial and informal for a scientific article. Please use the first sentence of the introduction to define Alzheimer's disease according to an official source (i.e. DSM-V, Alzheimer Association, National Institute on Aging etc).

Lines 32-34 "Although it is unknown what percentage of these diabetes people get AD, even if 10% of diabetic patients go on to develop AD, it would treble the number of AD patients worldwide [1]" - This is speculation without reference, please rephrase it to be less informal and speculative or add a reference supporting your claim.

Lines 36-38 "With age, the brain becomes more susceptible to cell damage caused by chronic hyperglycemia, and this can explain the onset of neuroinflammation and cognitive changes seen in some patients with diabetes" - speculation. Please provide a reference for this pathophysiologic mechanism

Lines 48-49 "insulin deficiency in the brain, by decreasing the permeability of the blood-brain, by decreasing the permeability of the blood-brain barrier to insulin" - this sentence needs to be rewritten

Lines 64-73 - everything here is speculation without references. Also, this part belongs to the discussion section of the paper, not the introduction.

Lines 79-81 "In the second step, they would reach the psychiatrist/neurologist in plenty of time, a condition in which the cognitive decline can be slowed down" - please add a reference that demonstrates successful slowing of cognitive decline with early neurologic/psychiatric interventions. 

Line 91 "and were being treated with OADs" - no full term for OAD, please add it

Line 92 "The participants were enrolled in the study during 2019-2022" - please add the exact months and "from x to y"

Line 109 "LDL, HDL" - abbreviations without full terms

Line 110 "HbA1c" - same as line 109

Lines 111-112 "With the exception of these, all the samples were tested on that day in the hospital laboratory" - rephrase for clarity

Line 115 "We use ELISA method." - full term for ELISA and rephrase to be grammatically correct.

Lines 121-122 "increased waist circumference , raised triglycerides, reduced HDL, elevated blood pressure and raised plasma glucose" - please add the exact cutoff values for each criterion

Line 128 "MMSE" - abbreviation

Line 144 "The MoCA took about ten minutes to administer" - please describe how the MoCa test is scored and interpreted

Line 147 "Statisstical analysis" - statistical.

Lines 160-161 "The majority of the patients were overweight (n=47, 31.8%), and obese (n=95, 64.2%)" - define the criteria for "overweight" and "obese"

Table 1 should have a glossary explaining each abbreviation at the bottom of the table

Lines 167-168 "we calculated the HOMA IR index. This was moderately increased in the study group, thus identifying patients with insulin resistance" - define the normal values of the HOMA-IR index

Lines 178-179 "More than a third of the patients in the study had microvascular complications, and almost one quarter of the patients had macrovascular complications (table 3)." - define microvascular and microvascular complications, respectively

Line 191 "MOCA scoreee" - typo

Lines 253-255 "There are rare longitudinal studies published who noted cognitive impairment in relation to metabolic syndrome elements (insulin resistnace)" - please add a reference for this

Lines 256-262 should be presented as a table of diagnostic criteria for the sake of legibility

Lines 267-270 "The authors calculated sensitivity and specificity, positive and negative predictive values, likelihood ratios, and the Kappa statistic.The MoCA appeared to be a better screening tool than the MMSE for MCI in the diabetic population" - please list the results that the authors obtained (sensitivity and specificitiy + area under ROC curve (AUC) if available).

Lines 297-299 "Diagnostic accuracy for MoCA and MMSE was calculated by ROC curves. The MoCA test better fulfills the criteria for screening tests to detect MCI in patients over 60 years of age than the MMSE" - please list the mentioned area under ROC curve values from the study

Lines 305-306 "her FBS (177.05 ± 62.48 vs. 149.38 ± 54.38) and PPBS (282.03 ± 85.61 vs. 04.61), which were statistically" - what are FBS and PPBS?

Reviewer 3 Report

This is an interesting study that has built on the work of previous researchers. Using a standardized test for the evaluation of cognition, the authors find cognitive decline is associated with glycemic control parameters and leptin level. However some minor concerns have been noted:

1. Include statistic analysis that was performed in the statistics section

2. Table 4 please use <0.001 instead of .000

3. Tables 5,6 add P sig column even though there are not significant difference

4. Tables 5,6,7 there is no label for the yes/no column. Is this diabetes complications?

5. p6. line 200 treatment is missing a "t"; p7 line 253 add abbreviation "mild cognitive impairment (MCI); p8 line 278 define 3MS

6. The discussion is a list of journal article reviews. Instead of listing the findings for each, how does your study support or add new insights to the field.

Round 2

Reviewer 2 Report

Dear authors,

I am pleased to notice that your manuscript has significantly improved in quality with the newest version.

However, I would kindly ask of you to read through it several times more, as I have noticed several spelling and formatting errors. 

Following correction of those, I am opined that your manuscript should be accepted for publication.

Author Response

Dear reviewer, 

Thank you for your cooperation and for the extremely useful information and advice you provided me. I have made the changes mentioned by you, and soon as I will have access to a computer, I will also send the manuscript. 

Thank you for your kind cooperation!